# The Impact of Different Patterns of Residual Disease on Long-Term Oncological Outcomes in Breast Cancer Patients Treated with Neo-Adjuvant Chemotherapy

**DOI:** 10.3390/cancers16020376

**Published:** 2024-01-16

**Authors:** Corrado Tinterri, Bethania Fernandes, Alberto Zambelli, Andrea Sagona, Erika Barbieri, Simone Di Maria Grimaldi, Shadya Sara Darwish, Flavia Jacobs, Camilla De Carlo, Martina Iuzzolino, Damiano Gentile

**Affiliations:** 1Breast Unit, IRCCS Humanitas Research Hospital, Via Manzoni 56, 20089 Rozzano, Milan, Italy; corrado.tinterri@hunimed.eu (C.T.); andrea.sagona@cancercenter.humanitas.it (A.S.); erika.barbieri@cancercenter.humanitas.it (E.B.); simone.dimariagrimaldi@cancercenter.humanitas.it (S.D.M.G.); shadya.darwish@humanitas.it (S.S.D.); 2Department of Biomedical Sciences, Humanitas University, Via Rita Levi Montalcini 4, 20090 Pieve Emanuele, Milan, Italy; alberto.zambelli@hunimed.eu (A.Z.); martina.iuzzolino@humanitas.it (M.I.); 3Department of Pathology, IRCCS Humanitas Research Hospital, Via Manzoni 56, 20089 Rozzano, Milan, Italy; bethania.fernandes@humanitas.it (B.F.); camilla.decarlo@humanitas.it (C.D.C.); 4Medical Oncology and Hematology Unit, IRCCS Humanitas Research Hospital, Via Manzoni 56, 20089 Rozzano, Milan, Italy; flavia.jacobs@humanitas.it

**Keywords:** breast cancer, neo-adjuvant chemotherapy, scattered, circumscribed, surgery

## Abstract

**Simple Summary:**

We investigated the predictive factors for different patterns of residual disease in the breast after neo-adjuvant chemotherapy and compared the long-term outcomes between the scattered versus the circumscribed pattern. A total of 219 histologic sections of postoperative surgical specimens were evaluated. Two independent predictive factors for the circumscribed pattern were identified: discontinuation of neo-adjuvant chemotherapy cycles and a tumor size >18 mm. Over a median follow-up of 74.7 months, disease-free survival and distant disease-free survival rates were similar for both patterns; however, the scattered pattern showed significantly better overall survival. Additionally, four independent factors associated with a higher risk of recurrence and worse survival outcomes were identified: discontinuation of neo-adjuvant chemotherapy cycles, tumor size >18 mm, triple negative breast cancer, and ypN+ disease. Our study highlights the importance of evaluating the patterns of residual disease for better postoperative management of breast cancer patients.

**Abstract:**

Backgrounds: The majority of breast cancer (BC) patients treated with neo-adjuvant chemotherapy (NAC) achieves a pathologic partial response with different patterns of residual disease. No clear correlation between these patterns and oncological results was described. Our aims were to define the predictive factors for different patterns of residual disease and compare the outcomes between the scattered versus the circumscribed pattern. Methods: We reviewed 219 postoperative surgical specimens. Patients were divided into two groups: scattered versus circumscribed. Disease-free survival (DFS), distant DFS (DDFS), and overall survival (OS) were analyzed. Results: The scattered and circumscribed patterns were assessed in 111 (50.7%) and 108 (49.3%) patients. Two independent predictive factors for the circumscribed pattern were identified: discontinuation of NAC cycles (*p* = 0.011), and tumor size post-NAC >18 mm (*p* = 0.022). No difference was observed in terms of DFS and DDFS. Patients with the scattered pattern exhibited a statistically significant better OS. Discontinuation of NAC cycles, tumor size >18 mm, triple-negative BC, and ypN+ were associated with increased recurrence and poorer survival. Conclusions: Discontinuation of NAC cycles and tumor size are independent factors associated with patterns of residual disease. The scattered pattern presents better survival. Understanding the relationship between NAC, the residual pattern, and differences in survival outcomes offers the potential to optimize the therapeutic approaches.

## 1. Introduction

Currently, neo-adjuvant chemotherapy (NAC) has become increasingly employed for the comprehensive care of breast cancer (BC) [1,2,3]. Since it has been demonstrated that NAC has similar survival outcomes compared with adjuvant chemotherapy, an increased number of BC patients has been treated with preoperative systemic therapy [4,5,6]. Neo-adjuvant chemotherapy is particularly beneficial for patients with locally advanced BC [7,8], triple negative BC, and human epidermal growth factor receptor 2 (HER2)-positive BC [9,10,11,12,13]. In the age of de-escalation surgery for BC, it has been shown in many studies that NAC increases the rate of breast conservative surgery (BCS) compared to mastectomy [14,15] and it leads to a de-escalation of axillary surgical management in selected cases [16,17]. Consequently, NAC has become a standard part of BC management and plays a crucial role in clinical decision-making. Moreover, the prognostic significance of pathologic complete response (pCR) has been deeply investigated and it is now widely known that achieving a pCR is significantly associated with improvements in recurrence and survival outcomes [18,19,20]. Cortazar P et al. [21] conducted a large meta-analysis including more than 11,000 BC patients which confirmed the assumption that there is a significant association between the extent of tumor response to NAC and long-term oncological results. However, the majority of BC patients undergoing NAC achieve a pathologic partial response (pPR) with different patterns of residual disease in the breast [22]. Additionally, the residual disease after NAC significantly correlates with the clinical prognosis [23]. Machine learning models have also been used in BC prognosis prediction [24,25]. Together these data highlight the clinical utility of evaluating the treatment response to NAC, not only for prognostic information but also for guiding therapeutic management in the era of increasingly tailored treatment strategies [26]. In this setting, different adjuvant therapies, tailored based on the biologic subtype of BC, have been used to improve survival outcomes. For example, the routine use of T-DM1 for HER2-positive BC with residual disease has been approved after the publication of the results of the KATHERINE trial [27]. Despite these advancements, a definitive method for evaluating posttreatment responses remains undeveloped and additional efforts are needed to develop a clinically meaningful risk stratification for BC patients with pPR after NAC, to better understand the different outcomes and refine treatment strategies further. While some studies have characterized the radiological patterns of residual disease in the breast [28,29,30], only two studies have focused on the histologic pattern of pPR, classifying them as either scattered or circumscribed [26,31]. Our research aims to identify predictive factors for these different patterns of residual disease in the breast after NAC and to compare the different long-term outcomes between the scattered versus the circumscribed pattern. 

## 2. Materials and Methods

### 2.1. Study Design

A detailed retrospective review of all the consecutive BC patients who presented pPR after NAC at the Breast Unit of IRCCS Humanitas Research Hospital (Milan, Italy) between October 2006 and April 2020 was performed. Patients with de novo metastatic disease were excluded from the present analysis. Among 304 cases identified, histologic sections of postoperative surgical specimens were available for review from 219 patients. Characteristics extracted from the institutional database and the medical charts included various patient characteristics: age, menopausal status, preoperative radiological staging, clinical tumor size and stage, nodal status, systemic therapy details (type and number of cycles), biologic subtype, histotype, vascular invasion presence, nodularity, pathologic tumor size and stage, type of surgery, and postoperative treatment. Hormone receptor status was assessed by immunohistochemistry (IHC). Estrogen and progesterone receptor status was considered positive if expressed in >1% immune-reactive cells. HER2 status was assessed by IHC (0, 1+, 2+, or 3+ score) and by fluorescent in situ hybridization (FISH), the latter performed in all patients with a HER2 IHC score of 2+. HER2 overexpression positivity was defined according to ASCO-CAP guidelines [32] for a membrane staining IHC score of 3+ or 2+ with evidence of FISH amplification. HER2 IHC scores of 1+ and 0 defined a HER2-negative status. Biologic subtype was defined based on the combination of hormone receptor status and HER2 status. A multidisciplinary tumor board, including breast surgeons, breast oncologists, breast pathologists, radiotherapists, oncoplastic surgeons, and radiologists gave indication to preoperative systemic therapy based on the tumor size, stage, and BC subtype and after the end of NAC discussed the surgical management of every patient. Discontinuation of NAC cycles was defined as to either prematurely ending the entire chemotherapy regimen or skipping one or more of the planned cycles. All patients underwent either BCS or mastectomy. Regarding axillary surgical management, all patients underwent either direct axillary lymph node dissection (ALND) or sentinel lymph node biopsy (SLNB); in the latter group, a subsequent ALND was performed if the sentinel lymph node was macrometastatic at intraoperative pathological evaluation. Hematoxylin and eosin-stained sections of the posttreatment surgical specimens were reviewed by three breast pathologists, blinded to the pretreatment clinical characteristics, to the histologic features and hormone receptor status, and to the oncological outcomes. They evaluated post-NAC tumor dimension, biologic subtype, histotype, and vascular invasion. The histologic patterns of response were evaluated based on the analysis performed by Pastorello R.G. et al. [31] and patients were divided into two different groups: scattered versus circumscribed patterns (Figure 1). The circumscribed pattern was defined either as a single focus of residual cancer within the tumor bed, or as small residual nests confined into a circumscribed area of the tumor bed. The scattered pattern, on the other hand, was described either as two or more foci of residual cancer within the tumor bed recognizable as distinct ones, or as small residual nests or single residual cells scattered across the tumor bed, within a broad area of treatment-related fibrosis. Each patient gave informed consent for operation and clinical data collection. 

### 2.2. Statistical Analysis

Descriptive statistics were used to characterize the overall cohort, with continuous variables reported as medians and ranges and categorical variables as frequencies and proportions. After central pathology review, clinic-pathological characteristics of the two different groups of patterns of residual disease in the breast were compared using the chi-square test. After that, a multivariate analysis was performed using a logistic regression model to identify independent predictors for patterns of residual disease. The multivariate analysis included any variable associated with the result in the univariate analysis (inclusion cutoff value *p* < 0.10). The Kaplan–Meier method was used to determine the recurrence and survival probabilities and the log-rank test was used to compare the two different groups of BC patients who presented pPR after NAC (scattered versus circumscribed pattern). Disease-free survival (DFS) was defined as the period from the date of surgical treatment to the date of any tumor progression including loco-regional recurrence or distant metastases. Distant disease-free survival (DDFS) was defined as the period from the date of surgery and the date of detection of distant metastases. Overall survival (OS) was defined as the time interval from surgical treatment to death from any cause or to the date of last contact. Additionally, multivariate analyses were performed using the Cox proportional hazards model to identify independent risk factors for DFS, DDFS, and OS. Hazard ratios (HR) and 95% confidence intervals (95%CI) were calculated. Statistical significance was set at *p* < 0.05; all statistical tests were two-tailed. The last follow-up was updated to 1 November 2023. Data analyses and figures were performed with IBM SPSS 25.0 software.

## 3. Results

### 3.1. Characteristics and Treatment of Breast Cancer Patients with Pathologic Partial Response after Neo-Adjuvant Chemotherapy

A total of 219 histologic sections of postoperative surgical specimens of BC patients who presented pPR after NAC were reviewed. The mean age of the patients was 50 years (range, 26–84), and 116 (53.0%) were postmenopausal. The majority of the patients (*n* = 140, 63.9%) underwent preoperative mammography, whereas breast magnetic resonance imaging was performed only in 65 (29.7%) patients. An ultrasound-guided biopsy of suspicious axillary lymph nodes was performed in 40 (18.3%) patients. The median size of the tumor pre-NAC was 33 mm (range, 12–115), and 132 (60.2%) patients presented a cT2 BC before NAC. The standard NAC treatment protocol, received by 157 (71.7%) patients, consisted of anthracycline and taxane (90 mg/m^2^ epirubicin or 60 mg/m^2^ doxorubicin combined with 600 mg/m^2^ cyclophosphamide every 3 weeks for four cycles followed by 75 mg/m^2^ docetaxel for four cycles or by 80 mg/m^2^ paclitaxel for twelve cycles). A total of 49 (22.4%) patients were treated solely with anthracycline-based NAC (FEC, EC or AC: 600 mg/m^2^ fluorouracil plus 600 mg/m^2^ cyclophosphamide plus 90 mg/m^2^ epirubicin or 60 mg/m^2^ doxorubicin, administered every 3 weeks for four cycles). Additionally, 52 (23.7%) and 8 (3.7%) HER2+ BC patients received either trastuzumab (loading dose of 8 mg/kg followed by 6 mg/kg in subsequent cycles) ± pertuzumab (loading dose of 840 mg followed by 420 mg in subsequent cycles) in combination with taxanes. Overall, 24 (11.0%) patients experienced discontinuation of NAC cycles before surgery. Luminal-like BCs were the most frequent biologic subtype (*n* = 114, 52.1%). The majority of the patients were treated with mastectomy (*n* = 130, 59.4%), and 141 (64.4%) patients underwent ALND either direct or following SLNB. The median size of the tumor post-NAC was 18 mm (range, 1–120), and 95 (43.4%) patients achieved ypN0 after NAC. Post-NAC treatments included capecitabine in 22 (10.1%) patients and T-DM1 in 50 (22.8%) patients. Regarding residual disease patterns, 111 (50.7%) patients exhibited a scattered pattern, whereas the circumscribed pattern was assessed in 108 (49.3%) patients. The overall population characteristics are detailed in Table 1.

### 3.2. Identification of Predictive Factors for Different Patterns of Residual Disease in the Breast after Neo-Adjuvant Chemotherapy

For further analysis, patients were divided into two different groups based on their pattern of residual disease in the breast after NAC: scattered versus circumscribed. Their clinic-pathologic characteristics were compared and multivariate analysis identified two independent predictive factors which were significantly associated with the circumscribed pattern: discontinuation of NAC cycles (scattered 4.5% versus circumscribed 17.6%, odds ratio (OR) = 0.255, 95%CI = 0.089–0.729, *p* = 0.011), and size of the tumor post-NAC > 18 mm (scattered 37.8% versus circumscribed 62.0%, OR = 2.013, 95%CI = 1.108–3.655, *p* = 0.022). The relationship between clinic-pathologic characteristics and patterns of residual disease is shown in Table 2.

### 3.3. Comparison of Long-Term Oncological Outcomes between Patients with Different Patterns of Residual Disease in the Breast after Neo-Adjuvant Chemotherapy and Independent Factors Influencing the Prognosis

After a median follow-up of 74.7 months (range, 44.3–182.6), 85 (38.8%) BC patients who presented pPR after NAC experienced a recurrence. Of these, 14 (/85, 16.5%) patients developed a loco-regional recurrence only, 51 (/85, 60.0%) patients developed metastases only, and 20 (/85, 23.5%) patients had both loco-regional and distant recurrences. A total of 37 (/111, 33.3%) and 48 (/108, 44.4%) patients developed a recurrence in the scattered and circumscribed group, respectively. There were 63 (28.8%) deaths in total; 23 (/111, 20.7%) and 40 (/108, 37.0%) patients in the scattered and circumscribed group died over the observation period, respectively. The DFS rate at 3, 5, and 10 years was 77.2%, 70.3%, 64.2%, and 65.4%, 59.3%, 54.2%, in the scattered and circumscribed groups, respectively. The DDFS rate at 3, 5, and 10 years was 81.7%, 73.9%, 64.6%, and 70.1%, 65.0%, 54.8%, in the scattered and circumscribed groups, respectively. The OS rate at 3, 5, and 10 years was 87.1%, 83.2%, 75.3%, and 84.1%, 72.1%, 61.8%, in the scattered and circumscribed groups, respectively. No difference was observed in terms of DFS and DDFS between the scattered and the circumscribed patterns (*p* = 0.117, *p* = 0.155, respectively); however, patients with the scattered pattern exhibited a statistically significant better OS compared with the circumscribed pattern (*p* = 0.022). Comparisons of long-term oncological outcomes between the scattered and circumscribed groups are summarized in Table 3 and Figure 2.

Four independent factors significantly associated with pPR patients’ risk of recurrence and survival were identified. Discontinuation of NAC cycles (HR = 2.846, 95%CI = 1.002–8.085, *p* = 0.050, HR = 3.181, 95%CI = 1.112–9.101, *p* = 0.031) and size of the tumor post-NAC >18 mm (HR = 2.691, 95%CI = 1.335–5.427, *p* = 0.006, HR = 3.130, 95%CI = 1.536–6.376, *p* = 0.002) were significantly associated with worse recurrence outcomes in terms of DFS and DDFS, respectively. Moreover, triple negative BC (HR = 14.645, 95%CI = 1.630–131.553, *p* = 0.017, HR = 12.063, 95%CI = 1.401–103.864, *p* = 0.023, HR = 29.146, 95%CI = 1.327–639.945, *p* = 0.032) and presence of residual disease in the axillary lymph nodes after NAC (HR = 3.566, 95%CI = 1.655–7.687, *p* = 0.001, HR = 3.873, 95%CI = 1.724–8.704, *p* = 0.001, HR = 2.565, 95%CI = 1.002–6.569, *p* = 0.050) were significantly associated with worse recurrence and survival outcomes in terms of DFS, DDFS, and OS, respectively. Independent risk factors related to recurrence and survival are summarized in Table 4.

## 4. Discussion

Our study focused on BC patients with pPR after NAC, specifically investigating the patterns of residual disease in the breast. Through a central pathology review, we showed that the identification of scattered and circumscribed residual disease patterns is clinically significant as these patterns serve as valuable prognostic indicators. After 74.7 months of median follow-up, DFS and DDFS rates were similar between the two patterns of residual disease; however, patients with the circumscribed pattern presented inferior OS.

Our multivariate analysis identified two independent predictive factors associated with the circumscribed pattern, including the discontinuation of NAC cycles and a post-NAC tumor size >18 mm. In a recent retrospective analysis, Pastorello R.G. et al. [31] compared the characteristics of 389 BC patients treated with NAC to different patterns of residual disease in the breast and showed that several BC features at presentation were significantly associated with the pattern of residual carcinoma among patients who did not achieve a pCR. They noted a significant association between the biologic subtype of the tumor and its pattern of response; in fact, among patients with hormone receptor positive/HER2-negative tumors, 89.4% had a scattered patter and only 10.6% had a circumscribed pattern. On the other hand, among patients with triple negative BC, 52.8% had a circumscribed pattern and 47.2% had a scattered pattern (*p* < 0.001). Additionally, lower tumor grade and larger tumor size pre-NAC were significantly associated with a scattered pattern (*p* = 0.002, *p* = 0.010, respectively).

The previous study did not provide an analysis of oncological results; however, in our analysis we showed that patients with the scattered pattern demonstrated a better OS compared to those with the circumscribed pattern. In contrast to our findings, Laws A. et al. [26] reported in their retrospective analysis that patients with a scattered residual tumor or minimal response had poorer recurrence-free survival outcomes compared to those with a concentric pattern, a trend that was also evident in terms of OS. After adjusted analyses for biologic subtype, ypN status, and pCR, both a scattered pattern and no/minimal response remained significantly associated with inferior recurrence-free survival and OS relative to a concentric pattern. Similarly, Chen A.M. et al. [33] performed a retrospective analysis on 340 BC patients treated with NAC followed by BCS and radiotherapy between 1987 and 2000 to determine patterns of loco-regional recurrence and ipsilateral BC recurrence. The residual tumor was characterized as multifocal or a solitary mass, which similarly correlated to the scattered or circumscribed patterns, respectively. Variables that positively correlated with loco-regional recurrence and ipsilateral BC recurrence were cN2-3 disease, post-NAC residual tumor >2 cm, lymphovascular space invasion, and a multifocal pattern of residual disease.

Note that beyond the binary pCR classification and the different patterns of residual disease, the MD Anderson Cancer Center developed the residual cancer burden (RCB) index as a method to quantify residual disease after NAC for BC [34]. The RCB index is based on histopathological variables such as number of involved nodes, size of the largest nodal metastasis, and size and percentage cellularity of the primary tumor bed. In subsequent studies, the RCB index proved to be a valid prognostic indicator for both DDFS and OS [35,36,37].

Moreover, in our study, we performed an analysis to evaluate independent factors for recurrence and survival, and we showed that the prognostic findings appeared to be driven by patients with discontinuation of NAC cycles and post-NAC tumor size >18 mm for any recurrence, as well as positive nodal status after NAC (ypN+) and triple negative BC for any recurrence and/or death. These findings suggest that the histologic pattern of residual disease in the breast may have implications beyond immediate recurrence and survival risk. Our results warrant further exploration since the specific pattern of residual disease may reflect a surrogate measure of treatment response, which would provide a biologic link to the observed association with OS. Understanding the factors contributing to this survival difference can inform targeted interventions and refine post-NAC management strategies.

While our study contributes valuable insights, it is essential to acknowledge its limitations. The retrospective nature of the analysis and the single-institution focus may introduce biases; in fact, the central pathologic evaluation of the different patterns of residual disease was performed by internal breast pathologists and reproducibility of this measure is needed before widespread adoption. Additionally, sections of postoperative surgical specimens of BC patients who presented pPR after NAC were not available for the entire cohort. Another limitation of our study is the absence of data on tumor grade in our database. Consequently, we were unable to evaluate its impact as a prognostic factor in our cohort of BC patients treated with NAC. This omission restricts our ability to fully assess the influence of tumor grade on long-term oncological outcomes.

## 5. Conclusions

In conclusion, our results indicate that the discontinuation of NAC cycles and the size of the tumor post-NAC are significant and independent factors associated with patterns of residual disease in the breast. Patients with a scattered pattern present better survival outcomes. Triple negative BC and ypN+ disease are significantly associated with worse recurrence and survival outcomes in patients with pPR after NAC. A better understanding of the relationship between preoperative chemotherapy, the residual pattern of disease in the breast, and the differences in survival outcomes may be of value in helping to guide postoperative systemic management for BC patients.

## Figures and Tables

**Figure 1 cancers-16-00376-f001:**
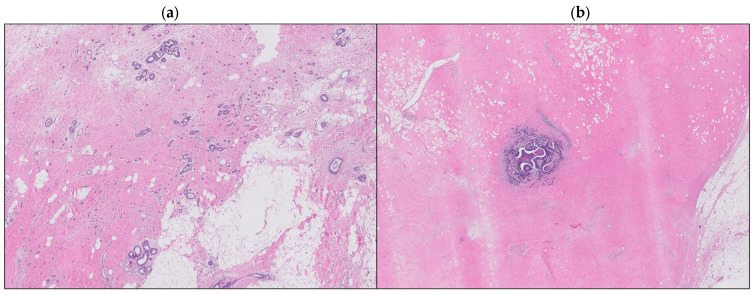
Scattered pattern (**a**), circumscribed pattern (**b**).

**Figure 2 cancers-16-00376-f002:**
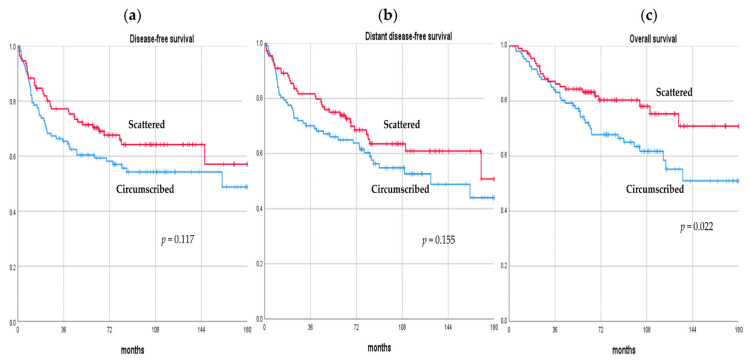
Disease-free survival (**a**), distant disease-free survival (**b**), and overall survival (**c**) curves of breast cancer patients with pathologic partial response after neo-adjuvant chemotherapy according to their pattern of residual disease.

**Table 1 cancers-16-00376-t001:** Characteristics of 219 patients with pathologic partial response after neo-adjuvant chemotherapy for breast cancer.

Characteristics	Number (%)/Median (Range)
Patients	
Age (years)	50 (26–84)
Postmenopausal	116 (53.0%)
Preoperative staging	
Mammography	140 (63.9%)
Breast and axillary US	495 (100%)
Axillary biopsy	40 (18.3%)
MRI	65 (29.7%)
PET	116 (53.0%)
Dimension pre-NAC (mm)	33 (12–115)
Stage pre-NAC	
cT1	20 (9.1%)
cT2	132 (60.2%)
cT3	39 (17.8%)
cT4	28 (12.9%)
cN0	73 (33.3%)
cN1	146 (66.7%)
NAC with anthracycline only	49 (22.4%)
NAC without anthracycline	13 (5.9%)
NAC with anthracycline and taxanes	157 (71.7%)
Trastuzumab	52 (23.7%)
Pertuzumab	8 (3.7%)
Complete NAC cycles	195 (89.0%)
Pattern of residual disease	
- Scattered	
- Circumscribed	111 (50.7%)
Tumor	108 (49.3%)
Subtype	
- Luminal-like	
- HER2-positive	114 (52.1%)
- Triple negative	53 (24.2%)
Histotype	52 (23.7%)
- Ductal	
- Lobular	195 (89.0%)
- Other	15 (6.9%)
Vascular invasion	9 (4.1%)
Single nodule	89 (40.6%)
Dimension post-NAC (mm)	168 (76.7%)
Stage post-NAC	18 (1–120)
- ypTmi	
- ypT1a	4 (1.8%)
- ypT1b	12 (5.5%)
- ypT1c	27 (12.3%)
- ypT2	62 (28.3%)
- ypT3	83 (37.9%)
- ypT4	20 (9.1%)
- ypN0	11 (5.1%)
- ypNmi	95 (43.4%)
- ypN1	5 (2.3%)
- ypN2	51 (23.3%)
- ypN3	41 (18.7%)
Surgical treatment	27 (12.3%)
- BCS	
- Mastectomy	89 (40.6%)
- SLNB not followed by ALND	130 (59.4%)
- SLNB followed by ALND	78 (35.6%)
- Direct ALND	28 (12.8%)
Postoperative treatment	113 (51.6%)
- Taxanes	
- Capecitabine	21 (9.6%)
- Radiotherapy	22 (10.1%)
- Endocrine	176 (80.4%)
- T-DM1	149 (68.0%)
	50 (22.8%)

US: ultrasound, MRI: magnetic resonance imaging, PET: positron emission tomography, NAC: neo-adjuvant chemotherapy, HER2: HER2 evaluated either on immunohistochemistry or on in situ hybridization, according to the ASCO CAP guidelines, BCS: breast-conserving surgery, SLNB: sentinel lymph node biopsy, ALND: axillary lymph node dissection, T-DM1: trastuzumab-emtansine.

**Table 2 cancers-16-00376-t002:** Correlation between clinic-pathological characteristics and of patterns of residual disease after neo-adjuvant chemotherapy for breast cancer.

Characteristics	Scattered (No. 111) Tot. (%)	Circumscribed (No. 108) Tot. (%)	Univariate Analysis	Multivariate Analysis
			*p*-Value	*p*-Value OR (95% CI)
Demographic				
Age (years)				
- ≤50	59 (53.2%)	56 (51.9%)	0.848	-
- >50	52 (46.8%)	52 (48.1%)	-	
Menopausal status				
- Pre-menopausal	54 (48.7%)	49 (45.4%)	0.629	-
- Postmenopausal	57 (51.3%)	59 (54.6%)	-	
Preoperative staging				
Dimension pre-NAC (mm)				
- ≤33	54 (48.7%)	50 (46.3%)	0.550	-
- >33	57 (51.3%)	58 (53.7%)	-	
Single nodule				
- Yes	89 (80.2%)	79 (73.2%)	0.220	-
- No	22 (19.8%)	29 (26.8%)	-	
Stage pre-NAC				
- cT1-2	80 (72.1%)	72 (66.7%)	0.388	-
- cT3-4	31 (27.9%)	36 (33.3%)	-	
NAC				
- Anthracycline and taxanes	76 (68.5%)	81 (75.0%)	0.399	-
- Anthracycline only	27 (24.3%)	22 (20.4%)	-	
- Without anthracycline	8 (7.2%)	5 (4.6%)	-	
Complete NAC cycles				
- Yes	106 (95.5%)	89 (82.4%)	0.002 ^a^	0.011 ^a^ 0.255 (0.089–0.729)
- No	5 (4.5%)	19 (17.6%)	-	-
Tumor				
Histotype				
- Ductal	98 (88.3%)	97 (89.8%)	0.106	-
- Lobular	10 (9.0%)	5 (4.6%)	-	
- Other	3 (2.7%)	6 (5.6%)	-	
Subtype				
- Luminal-like	58 (52.3%)	56 (51.9%)	0.362	-
- HER2-positive	32 (28.8%)	21 (19.4%)	-	
- Triple negative	21 (18.9%)	31 (28.7%)	-	
Dimension post-NAC (mm)				
- ≤18	69 (62.2%)	41 (38.0%)	<0.0001 ^a^	0.022 ^a^ 2.013 (1.108–3.655)
- >18	42 (37.8%)	67 (62.0%)	-	-
Stage post-NAC				
- ypT1-2	103 (92.8%)	85 (78.7%)	0.003 ^a^	0.074 2.328 (0.923–5.875)
- ypT3-4	8 (7.2%)	23 (21.3%)	-	-
Vascular invasion				
- Yes	45 (40.5%)	44 (40.7%)	0.976	-
- No	66 (59.5%)	64 (59.3%)	-	

OR: odds ratio, 95% CI: 95% confidence interval, NAC: neo-adjuvant chemotherapy, HER2: HER2 evaluated either on immunohistochemistry or on in situ hybridization, according to the ASCO CAP guidelines, ^a^: statistically significant.

**Table 3 cancers-16-00376-t003:** Comparison of disease-free, distant disease-free, and overall survival in patients with different patterns of residual disease after neo-adjuvant chemotherapy for breast cancer.

Outcomes	Scattered	Circumscribed	*p*-Value
DFS rate			
- 3-year	77.20%	65.40%	0.117
- 5-year	70.30%	59.30%	
- 10-year	64.20%	54.20%	
DDFS rate			
- 3-year	81.70%	70.10%	0.155
- 5-year	73.90%	65.00%	
- 10-year	64.60%	54.80%	
OS rate			
- 3-year	87.10%	84.10%	0.022 ^a^
- 5-year	83.20%	72.10%	
- 10-year	75.30%	61.80%	

DFS: disease-free survival, DDFS: distant disease-free survival, OS: overall survival, ^a^: statistically significant.

**Table 4 cancers-16-00376-t004:** Multivariate analysis of independent factors influencing the prognosis of patients with pathologic partial response after neo-adjuvant chemotherapy for breast cancer.

Independent Factors	DFS	DDFS	OS
	HR (95%CI) *p*-Value	HR (95%CI) *p*-Value	HR (95%CI) *p*-Value
Patient			
Age (years)			
- ≤50	Reference	Reference	Reference
- >50	0.787 (0.304–2.035) 0.621	0.622 (0.234–1.649) 0.340	1.401 (0.379–5.172) 0.613
Menopausal status			
- Pre-menopausal	Reference	Reference	Reference
- Postmenopausal	1.204 (0.472–3.072) 0.697	1.264 (0.491–3.255) 0.627	0.732 (0.212–2.524) 0.622
Preoperative staging			
Dimension pre-NAC (mm)			
- ≤33	Reference	Reference	Reference
- >33	0.733 (0.355–1.516) 0.402	0.642 (0.322–1.282) 0.209	0.570 (0.244–1.331) 0.194
Single nodule			
- Yes	Reference	Reference	Reference
- No	0.729 (0.348–1.527) 0.402	0.812 (0.383–1.722) 0.586	0.692 (0.275–1.737) 0.433
Stage pre-NAC			
- cT1-2	Reference	Reference	Reference
- cT3-4	1.507 (0.676–3.360) 0.316	1.467 (0.650–3.311) 0.356	0.709 (0.256–1.966) 0.509
NAC			
- With anthracycline	Reference	Reference	Reference
- Without anthracycline	0.829 (0.594–1.157) 0.270	0.929 (0.666–1.298) 0.667	0.876 (0.593–1.294) 0.506
Complete NAC cycles			
- Yes	Reference	Reference	Reference
- No	2.846 (1.002–8.085) 0.050 ^a^	3.181 (1.112–9.101) 0.031 ^a^	1.800 (0.555–55) 0.327
Tumor			
Histotype			
- Ductal	Reference	Reference	Reference
- Other	0.941 (0.359–2.468) 0.902	1.003 (0.380–2.651) 0.995	1.523 (0.518–4.479) 0.445
Pattern of response			
- Scattered	Reference	Reference	Reference
- Circumscribed	1.423 (0.741–2.732) 0.290	1.302 (0.684–2.482) 0.422	1.410 (0.601–3.307) 0.429
Subtype			
- HR+HER−	Reference	Reference	Reference
- Other	1.892 (0.381–9.411) 0.436	2.113 (0.426–10.484) 0.360	2.360 (0.221–25.177) 0.477
- Triple negative	Reference	Reference	Reference
- Other	14.645 (1.630–131.553) 0.017 ^a^	12.063 (1.401–103.864) 0.023 ^a^	29.146 (1.327–639.945) 0.032 ^a^
Dimension post-NAC (mm)			
- ≤18	Reference	Reference	Reference
- >18	2.691 (1.335–5.427) 0.006 ^a^	3.130 (1.536–6.376) 0.002 ^a^	2.159 (0.892–5.224) 0.088
Stage post-NAC			
- ypT1-2	Reference	Reference	Reference
- ypT3-4	0.930 (0.393–2.198) 0.868	0.990 (0.417–2.352) 0.982	2.424 (0.911–6.448) 0.076
- ypN0	Reference	Reference	Reference
- ypN+	3.566 (1.655–7.687) 0.001 ^a^	3.873 (1.724–8.704) 0.001 ^a^	2.565 (1.002–6.569) 0.050 ^a^
Vascular invasion			
- Yes	Reference	Reference	Reference
- No	1.660 (0.889–3.101) 0.112	1.625 (0.849–3.112) 0.143	2.134 (0.973–4.679) 0.058
Treatment			
Operation			
- BCS	Reference	Reference	Reference
- Mastectomy	1.101 (0.539–2.249) 0.793	1.108 (0.531–2.314) 0.784	1.265 (0.509–3.142) 0.612
Adjuvant radiotherapy			
- Yes	Reference	Reference	Reference
- No	1.111 (0.499–2.475) 0.796	1.045 (0.470–2.323) 0.913	1.110 (0.410–2.952) 0.796
Adjuvant chemotherapy			
- Yes	Reference	Reference	Reference
- No	1.370 (0.701–2.677) 0.357	1.257 (0.633–2.497) 0.514	1.271 (0.563–2.870) 0.564
Endocrine therapy			
- Yes	Reference	Reference	Reference
- No	1.385 (0.304–5.109) 0.608	1.278 (0.345–4736) 0.713	2.188 (0.259–18.477) 0.472
T-DM1			
- Yes	Reference	Reference	Reference
- No	2.703 (0.531–13.764) 0.231	2.409 (0.488–11.901) 0.281	1.574 (0.173–14.327) 0.687

DFS: disease-free survival, DDFS: distant disease-free survival, OS: overall survival, HR: hazard ratio, 95%CI: 95% confidence interval, NAC: neo-adjuvant chemotherapy, HR: hormone receptor, HER2: HER2 evaluated either on immunohistochemistry or on in situ hybridization, according to the ASCO CAP guidelines, BCS: breast-conserving surgery, T-DM1: trastuzumab-emtansine, ^a^: statistically significant.

## Data Availability

Data supporting reported results can be found in Appendix A.

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
