# Peer review of "The Impact of Different Patterns of Residual Disease on Long-Term Oncological Outcomes in Breast Cancer Patients Treated with Neo-Adjuvant Chemotherapy"

_cancers, 2024, doi:10.3390/cancers16020376_

Round 1
Reviewer 1 Report
Comments and Suggestions for Authors
Dear authors,
I read with interest the article which highlights the importance of evaluating the patterns of residual disease for better post-operative management of breast cancer patients.
I agree with the authors that a better understanding of the relationship between pre-operative chemotherapy, the residual pattern in the breast, and the differences in survival outcomes may be of value to guide post-operative systemic management for BC patients.
However the reproducibility of this measure measure is needed before widespread adoption.
I would like to suggest that the authors better explain the definition adopted for "scattered / circumscribed" in order to make it more reproducible in further clinical studies.
I recommend this article with these minor reviews.
The manuscript clear, relevant for the field and presented in a well-structured manner.
The cited references mostly recent publications (within the last 5 years) are relevant.
The manuscript scientifically sound and is the experimental design appropriate to test the hypothesis.
The figures/tables/images/schemes are appropriate and they properly show the data (easy to interpret and understand).
The statistical analysis or data acquired are appropriate.
The conclusions are consistent with the evidence and arguments presented the ethics statements and data availability statements to ensure they are adequate.
The results interpreted appropriately and are significant.
The article written in an appropriate way and the data and analyses presented appropriately.
Author Response
We thank the reviewer for the comment.
The materials and methods section was modified accordingly and the definition adopted for scattered/circumscribed pattern was expanded.
Reviewer 2 Report
Comments and Suggestions for Authors
This study focused on breast cancer (BC) patients who achieved a pathological complete response (pPR) after neoadjuvant chemotherapy (NAC), specifically examining patterns of residual disease in the breast. Through central pathology review, the identification of scattered and circumscribed residual disease patterns was found to be clinically significant, serving as valuable prognostic indicators. After a median follow-up of 74.7 months, disease-free survival (DFS) and distant disease-free survival (DDFS) rates were similar between the two patterns, but patients with the circumscribed pattern exhibited inferior overall survival (OS). Two independent predictive factors for the circumscribed pattern were identified: discontinuation of NAC cycles and a post-NAC tumor size >18 mm.
Comparisons with previous studies revealed associations between biological subtypes and patterns of response. Notably, patients with hormone receptor-positive/HER2-negative tumors tended to have a scattered pattern, while triple-negative BC patients were more likely to exhibit a circumscribed pattern. The study also demonstrated that patients with the scattered pattern had better OS compared to those with the circumscribed pattern, differing from some previous findings. Other studies have reported conflicting results regarding recurrence-free survival outcomes based on residual tumor patterns.
The analysis explored independent factors for recurrence and survival, revealing that discontinuation of NAC cycles and post-NAC tumor size >18 mm were associated with any recurrence, as well as positive nodal status after NAC (ypN+) and triple-negative BC were linked to recurrence and/or death. The study suggests that the histologic pattern of residual disease may extend beyond immediate recurrence and survival risk, possibly serving as a surrogate measure of treatment response.
While the study provides valuable insights, it acknowledges limitations, including its retrospective nature and single-institution focus. The need for reproducibility in central pathologic evaluation and the unavailability of post-operative surgical specimens for the entire cohort are recognized challenges. Despite these limitations, the study's findings prompt further exploration, indicating that understanding the factors contributing to survival differences based on residual disease patterns can inform targeted interventions and refine post-NAC management strategies.
Comments on the Quality of English LanguageThe text is well-crafted, and a few slight enhancements can be made.
Author Response
We thank the reviewer for the comments.
Reviewer 3 Report
Comments and Suggestions for Authors
This eminent retrospective study shows a novel approach to estimate Overall Survival (OS) in breast cancer. The Methods are sound, and the Results show clear tables that support the Discussion. I have some remarks:
Methods:
Why was the Chi Square test (apparently without correction) chosen over the standard Fisher Exact test?
Results:
Table 2 shows a surprisingly low ratio of lobular carcinoma, as compared to ductal carcinoma; do you have an explanation for this?
Discussion:
You should elaborate on tumor grade, as it is hardly touched in this article (as compared to tumor stage).
Author Response
This eminent retrospective study shows a novel approach to estimate Overall Survival (OS) in breast cancer. The Methods are sound, and the Results show clear tables that support the Discussion. I have some remarks:
Methods:
Why was the Chi Square test (apparently without correction) chosen over the standard Fisher Exact test?
Reply: We thank the reviewer for the comment.
The Chi Square test was used for categorical variables. Given the specific context of the study, where 219 BC patients were analyzed and considering the likely distribution of data across various subgroups, the Chi Square test was considered more suitable.
Results:
Table 2 shows a surprisingly low ratio of lobular carcinoma, as compared to ductal carcinoma; do you have an explanation for this?
Reply: We thank the reviewer for the insightful observation.
Upon re-evaluating our data in light of your comment, we found that the proportion of lobular carcinoma in our study cohort aligns closely with our institutional historical data, so unfortunately we do not have an explanation for this.
Discussion:
You should elaborate on tumor grade, as it is hardly touched in this article (as compared to tumor stage).
Reply: We thank the reviewer for the comment.
We appreciate your suggestion to discuss the role of tumor grade in our study. However, we regret to inform you that data on tumor grade in our institutional database was largely missing (especially for older cases) and it has been decided to exclude this tumor feature from the analysis of the patient cohort under consideration. This limitation prevented us from evaluating and analyzing the impact of tumor grade as a prognostic factor in our population of BC patients treated with NAC. We recognize the importance of tumor grade in BC prognosis and this aspect is now highlighted in the Limitations section.
Reviewer 4 Report
Comments and Suggestions for Authors The article is an original report presenting the results of retrospective study in breast cancer patients with residual tumors after neoadjuvant chemotherapy. The study compared outcomes between scattered vs circumscribed pattern. This topic is interesting and developing. The authors concluded that the scattered pattern presented better survival. I would suggest some changes: 1. In abstract add for “tumor size>18 mm” that it is size in the tumor after NAC 2. Please change in section “Results” that patients were treated with trastuzumab ± pertuzumab – pertuzumab is not used as monotherapy, always as dual blocade 3. Please clarify the threshold of 33 mm pre-NAC dimension in table 2. 4. Please give the definition of discontinuation of NAC – one cycle skipped was enough to be in this group or other criterium? Overall, the topic is important and appropriate for “Cancers” Journal and its audience. The article gives the reader interesting information about new aspects of residual disease after NAC in breast cancer. I recommend the article for publication in “Cancers” after minor revision. Comments on the Quality of English Languageminor
Author Response
The article is an original report presenting the results of retrospective study in breast cancer patients with residual tumors after neoadjuvant chemotherapy. The study compared outcomes between scattered vs circumscribed pattern. This topic is interesting and developing. The authors concluded that the scattered pattern presented better survival. I would suggest some changes:
- In abstract add for “tumor size>18 mm” that it is size in the tumor after NAC
Reply: We thank the reviewer for the comment.
The abstract was modified accordingly.
- Please change in section “Results” that patients were treated with trastuzumab ± pertuzumab – pertuzumab is not used as monotherapy, always as dual blocade
Reply: We thank the reviewer for the comment.
The results section was modified accordingly.
- Please clarify the threshold of 33 mm pre-NAC dimension in table 2.
Reply: We thank the reviewer for the comment.
The threshold was chosen because the median size of the tumor pre-NAC was 33 mm.
- Please give the definition of discontinuation of NAC – one cycle skipped was enough to be in this group or other criterium?
Reply: We thank the reviewer for the comment.
Discontinuation of NAC cycles was defined as to either prematurely ending the entire chemotherapy regimen or skipping one or more of the planned cycles. The materials and methods section was modified accordingly.
Overall, the topic is important and appropriate for “Cancers” Journal and its audience. The article gives the reader interesting information about new aspects of residual disease after NAC in breast cancer. I recommend the article for publication in “Cancers” after minor revision.
Reviewer 5 Report
Comments and Suggestions for Authors
The study explored factors influencing different patterns of residual breast disease after neo-adjuvant chemotherapy. They found that discontinuing chemotherapy cycles and tumor size >18 mm were linked to a circumscribed pattern. The methods are properly implemented, and the manuscript is well-written. However, I have some minor suggestions/concerns:
- The introduction apart is relatively small to cover the whole area of oncology for guiding the treatment of breast cancer. I suggest to add background section. Highlight the mentioned methods more, and also include some recent model including: PMID: 30972106 and/or PMID: 35205681 and/or similar models.
- The value p <0.10 in line 124 is quite large, is it a typo, or the authors have tested 0.05 or less and did not work?
- If possible for Fig 2 (KM plots), Can the authors draw the CI at each point that shows the interval areas around each curve, this may show some difference at some points!
-
Author Response
The study explored factors influencing different patterns of residual breast disease after neo-adjuvant chemotherapy. They found that discontinuing chemotherapy cycles and tumor size >18 mm were linked to a circumscribed pattern. The methods are properly implemented, and the manuscript is well-written. However, I have some minor suggestions/concerns:
- The introduction apart is relatively small to cover the whole area of oncology for guiding the treatment of breast cancer. I suggest to add background section. Highlight the mentioned methods more, and also include some recent model including: PMID: 30972106 and/or PMID: 35205681 and/or similar models.
Reply: We thank the reviewer for the comment.
The introduction section was modified accordingly.
- The value p <0.10 in line 124 is quite large, is it a typo, or the authors have tested 0.05 or less and did not work?
Reply: Thank you for your observation regarding the inclusion cut-off value of p<0.10 in our multivariate analysis. We would like to clarify that this was not a typographical error, but a deliberate methodological choice. The rationale was to reduce the risk of prematurely excluding variables that could have meaningful impact on the model. It is important to note that a higher p-value threshold in the initial model-building phase is not intended to suggest weaker evidence but rather serves as a means to capture a broader range of potentially influential variables.
- If possible for Fig 2 (KM plots), Can the authors draw the CI at each point that shows the interval areas around each curve, this may show some difference at some points!
Reply: Thank you for your valuable suggestion regarding the inclusion of CIs at specific points on the Kaplan-Meier plots in Figure 2 of our manuscript. After careful consideration and review of our data visualization capabilities, we find that adding CIs at specific points on the Kaplan-Meier curves would significantly increase the complexity of the figure. Our concern is that such detailed additions might potentially overcomplicate the visual presentation, making it more challenging for readers to interpret the key findings effectively. We hope that our decision to maintain the current format of Figure 2, prioritizing clarity and readability, aligns with the overall aim of effectively conveying our study's findings. We are grateful for your understanding and for your insightful feedback, which has prompted thoughtful consideration of how best to present our data.